# A Call for Action on Chronic Respiratory Diseases within Physical Activity Policies, Guidelines and Action Plans: Let’s Move!

**DOI:** 10.3390/ijerph192416986

**Published:** 2022-12-17

**Authors:** Mark W. Orme, Akila R. Jayamaha, Lais Santin, Sally J. Singh, Fabio Pitta

**Affiliations:** 1Department of Respiratory Sciences, University of Leicester, Leicester LE1 7RH, UK; 2Centre for Exercise and Rehabilitation Science, NIHR Leicester Biomedical Research Centre-Respiratory, University Hospitals of Leicester NHS Trust, Leicester LE3 9QP, UK; 3Department of Research and Development, Faculty of Nursing, KAATSU International University, Battaramulla 10120, Sri Lanka; 4Laboratory of Research in Respiratory Physiotherapy, Health Sciences Center, Universidade Estadual de Londrina, Londrina 86057-970, Brazil

**Keywords:** physical activity, sedentary behavior, guidelines, recommendations, health policy, chronic respiratory diseases, low- and middle-income countries

## Abstract

Global policy documents for the promotion of physical activity (PA) play an important role in the measurement, evaluation, and monitoring of population PA levels. The World Health Organisation (WHO) guidelines include, for the first time, recommendations for specific populations, including individuals living with a range of non-communicable diseases. Of note, is the absence of any chronic respiratory diseases (CRDs) within the recommendations. Globally, CRDs are highly prevalent, are attributable to significant individual and societal burdens, and are characterised by low PA. As a community, there is a need to come together to understand how to increase CRD representation within global PA policy documents, including where the evidence gaps are and how we can align with PA research in other contexts. In this commentary, the potential for synergy between evidence into the relationships between PA in CRDs globally and the relevance to current policies, guidelines and action plans on population levels of PA are discussed. Furthermore, actions and considerations for future research, including the need to harmonize and promote PA assessment (particularly in low- and middle-income countries) and encompass the synergistic influences of PA, sedentary behaviour and sleep on health outcomes in CRD populations are presented.

## 1. Introduction

There has been a growth in the number of global physical activity (PA) policy documents over the last two decades, in parallel with the amount (and quality) of evidence linking physical inactivity to non-communicable diseases (NCDs) [1]. The World Health Organisation (WHO) leads global PA policy, releasing the latest PA guidelines in 2020 [2]. The 2020 WHO guidelines reaffirm messages that some PA is better than none and that more PA confers additional health benefits [2]. They also highlight the importance of skeletal muscle-strengthening activities. In keeping with ‘something is better than nothing’, the 2020 WHO guidelines recommend, for the first time, reducing sedentary behaviour (SB) [2]. Another ‘first’ included within these guidelines are specific, although brief, recommendations for special populations (including individuals living with NCDs) [2].

The absence of chronic respiratory diseases (CRDs) within the WHO special recommendations is noteworthy given the prevalence, individual and societal burden, and low PA (and high SB) of these populations. Globally, CRDs are some of the most prevalent NCDs [3], accounting for 3.9 million annual deaths [4]. The PA levels of people living with CRDs are not only lower than healthy adults [5,6,7,8,9], but also compared with individuals living with other NCDs [10]. Low levels of PA for people living with chronic obstructive pulmonary disease (COPD) are associated with an increased risk of hospitalisation [11] and premature mortality [12]. Despite their high prevalence and burden, CRDs have received less public attention and less research funding globally compared with other NCDs [13,14].

It is important to note that the WHO guidelines are based on the consensus of the most recent evidence for the health impacts of PA and SB [2]. Evidence from the following 10 NCDs and disabilities were included: attention deficit hyperactivity disorder, hypertension, intellectual disability, major clinical depression, muscular sclerosis, Parkinson’s disease, schizophrenia, spinal cord injury, stroke survivors and type 2 diabetes [2]. CRDs were not included and comprise ~12% of the world’s population [15]. Although the need to conduct more population-based studies on people living with NCDs has been acknowledged [16], a concerted effort is needed to establish a sufficient evidence base for the benefits and recommendations of PA and SB for people living with CRDs. 

As a community, we need to come together to understand how we can increase CRD representation within global PA policy documents, including where the evidence gaps are and how we can align with PA research in other contexts. Undoubtedly, research into PA in CRD populations has already contributed to the understanding of how to increase population levels of PA and has still much to contribute. 

## 2. Multimorbidity

Going forward, guidelines and the promotion of PA for special populations will likely need to consider the growing high prevalence and interplay of comorbidities and multimorbidity [17,18,19]. Globally, an estimated 33% of adults live with at least two NCDs [20] and CRD populations represent a group of individuals experiencing a particularly high prevalence of comorbidities, including depression, metabolic disorders and cardiovascular disease [21] which are represented in the WHO guidelines. In COPD, the majority of individuals have at least one additional condition and almost half have more than two conditions [22]. Comorbidities in CRDs contribute to reduced physical functioning [23], poor health-related quality of life [24] and increased mortality [25,26,27]. Experiencing concomitant health conditions is a characteristic shared by many CRDs [28] which has also been linked to lower levels of PA [29]. The beneficial impacts of PA in 35 NCDs demonstrates the potential of PA for CRD populations and in the context of multimorbidity [30,31,32].

## 3. Lack of Data from LMIC

The reliance on evidence from high-income countries (HIC) to develop WHO PA guidelines is acknowledged. Research into PA of CRD populations in low- and middle-income countries (LMIC) can play an important role in public health guidance. With more than 90% of CRD-related deaths occurring in LMIC [33] but a lack of available data on PA in these countries [34], means an additional priority must be to boost research activities in this area. The combination of a lack of data, the different contexts in which PA is conducted [35,36] and the contrasting priorities emphasise the need for significant investment in PA research in LMIC [37], and specifically CRDs within LMIC.

The lack of available data on PA and SB for CRD populations in LMIC has been highlighted by a systematic review of published articles until January 2020 [34]. Of only 89 included articles, none were from low-income countries and 80% were conducted in upper-middle-income countries [34]. More than half of the included studies were from the Region of the Americas (51%), with 48% of articles from Brazil. Within other regions, evidence was also predominantly from a single country (e.g., 8/11 articles in the European region were from Turkey; 6/10 articles in South-East Asia were from India; all three articles in Africa were from Nigeria) [34]. Overall, two-thirds of the included articles originated from 4 out of 136 LMIC.

## 4. Appropriateness of Global Policies, Guidelines and Action Plans to CRDs

Increasing global levels of PA is not only important for health but would also directly contribute to several of the United Nations Sustainable Development Goals (SDGs) [38]; acknowledged by the WHO within their Global Action Plan on PA (GAPPA) [39]. Targets for countries to reduce physical inactivity by 10% and 15% by 2025 and 2030, respectively, have been set out by the GAPPA [39]. In conjunction with this, the International Society for Physical Activity and Health (ISPAH) set out the eight investments that work for PA [40]. These eight investments emphasize the need for a systems-based approach to tackle the current pandemic of physical inactivity, comprising: whole-of-school programmes, active transport, active urban design, healthcare, public education, sport and recreation for all, workplaces, and community-wide programmes [40].

When considering approaches to increase PA levels of CRD populations globally, ISPAH’s eight investments for PA can offer a helpful framework [37]. Across all eight investments, critical learning and potential benefit to people living with CRD can be obtained. Some examples of how these eight investments may interact and relate to CRD populations are outlined in Table 1.

## 5. A Brief History of PA Measurement in CRDs

Device-based assessments of PA and SB are now commonplace within CRD research, but there remain significant challenges to bring this field in line with wider global PA communities. While a range of devices were being discussed [41], early studies assessing PA (and SB) in CRD populations used the Dynaport Activity Monitor (DAM, McRoberts, The Hague, The Netherlands) [42,43,44] and/or the SenseWear Pro Armband (SWA, BodyMedia Inc., Pittsburgh, PA, USA) [45]. The popularity of DAM and SWA devices within CRD PA research is evident from systematic reviews describing the ways in which PA and SB have been assessed in COPD (largely from HIC) [46,47] and CRDs within LMIC [34]. From research conducted in LMIC, different models of DAM were used in 58% of studies using accelerometers (and in 42% of predominantly HIC studies [47]), with SWA used in 27% and simultaneous deployment of DAM and SWA in 19% of studies using accelerometers [34]. More recently, the SWA has been discontinued [48]. In recent years, the ActiGraph activity monitors (ActiGraph LLC, Pensacola, FL, USA) have been commonly used for PA assessment in individuals with COPD [49] and other populations [46]. Many other PA monitors are available, with wide variation in the depth of their validity in COPD and CRDs.

In line with technological advancements, accelerometer-based PA has been made more widely available and is now commonplace in large population (non-CRD) surveillance studies [50,51,52,53,54,55]. As advancements in accelerometer-based PA measurement continued, comparatively newer devices such as the GeneActiv and Axivity have also become widely used, including in large epidemiological cohorts (e.g., Doherty et al., 2017; Ricardo et al., 2020 [54,56]), but still largely underutilised in CRD populations [57]. 

There is an opportunity for the CRD PA community to align more closely with developments in the measurement of PA in other populations/contexts. There has been a shift towards more recognized devices from groups such as the Patient-Reported Outcome in COPD project (PROactive) consortium [58] and an International TaskForce on Physical Activity has been set up specifically in the context of CRDs [59]; initiatives which align well with this call for action. 

## 6. Actions and Considerations for Future Research

### 6.1. Harmonizing Device-Based Physical Activity Assessment, Especially in the Context of Low- and Middle-Income Countries

A fundamental challenge remains in increasing accessibility to PA monitors to assess physical inactivity and sedentary behaviour, especially for their use in large trials and assessments in clinical practice. Another important challenge lies in how best to process and analyse accelerometry data to generate clinically meaningful PA outcomes. This challenge is particularly highlighted in CRD literature, with accelerometer-based PA variables almost exclusively device-specific [34], preventing comparison between different devices and between studies globally, although this limitation is steadily being overcome. When considering PA intensity for CRD populations, another challenge is that the thresholds used to denote intensity (e.g., sedentary, light, moderate, vigorous) are derived from calibration studies with younger, healthier adults with relatively preserved exercise capacity [53,60]. Despite this, these PA intensity thresholds are commonly applied to people living with CRDs [47]. This may also have been driven by the need to translate PA data in clinically meaningful ways (e.g., step counts, time spent walking or in moderate-to-vigorous physical activity (MVPA), time spent sedentary or adherence to guidelines). However, more robust and precise measurement is needed when examining health associations, intervention effectiveness and population surveillance [61]. For example, it may be helpful to examine PA intensity in relation to tests of exercise capacity [57]. If we are to advocate for CRD populations to be included in global health PA policies, guidelines and action plans, then we must adopt more sophisticated analytical processes that can be harmonised with other international research. 

A crucial advancement from accelerometers such as the SWA (hindered by and limited to proprietary algorithms) to the more recent generation of devices is the measurement of acceleration directly in the International System of Units (e.g., gravitational units). Consequently, it becomes possible to generate harmonised (retrospectively or prospectively) large databases of comparable PA data, irrespective of specific device selection and post-processing decisions. Retrospective harmonisation of existing PA data in CRD populations is currently quite limited. If the latest generation of accelerometers is further adopted, then it may become possible to pool PA data into a large international database for people living with CRDs. Similar examples already exist, such as in children [62] and, more recently, in LMIC [63]. Such an initiative would offer considerable scientific benefit, including comparable cross-country prevalence estimates of physical inactivity, investigations into dose-response relationships between PA and health outcomes and examining determinants of PA in CRDs. Indeed, open-source resources such as GGIR [64] can be used to process and analyse acceleration data [65,66]. As advancements in accelerometer data processing and analysis continue, the use of these devices and analytical approaches have started to be implemented in large epidemiological studies. However, this transition is largely in HIC [67]. In LMIC, self-report measures of PA remain the predominant method of assessment, especially in the context of CRDs [34]. 

### 6.2. Harmonizing Questionnaire-Based Physical Activity Assessment

In resource-limited settings, whether in LMIC or HIC, self-reported measures of PA remain important data collection options. In CRD studies conducted in LMIC, the International Physical Activity Questionnaire (IPAQ) was the most common questionnaire used [34], aligning with global usage data which, as of 2016, reported the IPAQ being used in more than 50 countries [68]. Of note, the IPAQ has some limitations which impact its potential as a candidate for a globally harmonised approach for PA assessment for CRD populations. The IPAQ was developed for people aged 15–69 years (not comparable with the demographics of many CRDs) [69] and does not capture the context in which PA is undertaken (which can be detrimental to recall). To address this, the Global Physical Activity Questionnaire (GPAQ) was developed by adapting the IPAQ to incorporate work, leisure and transport contexts [70]. As of 2016, the GPAQ has been used in more than 100 countries [68]. Whilst the IPAQ and GPAQ both allow comparisons with PA guidelines, issues of compatibility, accuracy at the individual level and adoption of these questionnaires within CRD research limit their current potential as a tool for PA harmonisation.

### 6.3. More Comprehensive Analysis of Activity Behaviours: Encompassing Physical Activity, Sedentary Behaviour and Sleep

When we consider PA, we cannot do so in isolation from other behaviours that comprise our everyday lives. Specifically, SB and sleep make up the remaining 24 h each day, and by focusing only on physical activity, we risk missing the bigger slices of the pie, which would help to develop more tailored interventions. PA, SB and sleep are not only distinct behaviours that impact health, but they also interact with each other to impact health) [71,72]. Despite this, these 24 h physical behaviours are often examined and interpreted in isolation [34]. 

In CRD research, few studies are exploring the relationships between PA, SB and/or sleep on health outcomes [8,73,74,75,76] or response to interventions [77,78]. Given the low levels of PA, high SB and poor sleep quality in CRD populations, it is important to consider how these behaviours combine to influence health. Therefore, interventions that increase PA and reduce SB, including those aimed at behaviour modification, are indicated to people living with CRDs [79]. For example, in an adult population, Ekelund and colleagues showed that individuals with high sedentary time (11 h/day) required 40 min/day of MVPA for a 30% all-cause mortality risk reduction compared with 5 min/day of MVPA for people with lower levels of SB (6 h/day) for the equivalent risk reduction [71]. In COPD, time spent in sedentary behaviour above 8 h and 30 min per day has been strongly linked to a higher risk of mortality, even after adjusting for MVPA and a number of other variables [74]. Also in COPD, objectively measured better sleep quality-quantity has been associated with being more physically active [75,76]. A better understanding of the proportion of time distributed among PA, SB and sleep, such as through compositional data analysis, may be helpful in optimising interventions to improve health [73]. By establishing an evidence base that examines the combined effects (or ‘balance’) of these time-based behaviours, it becomes possible to tailor advice to individuals based on their capabilities and social circumstance [72], which is particularly applicable within CRD populations. 

## 7. Summary and Next Steps

It is important to increase the representation of CRDs within global policy documents of PA and SB, mainly focusing on where the gaps are and how we can align them with current and future best research practices. The impact of multimorbidities and scarce data from LMIC must be addressed for CRDs to be included in future global PA guidelines. A number of investments are outlined, which emphasize the need for a systems-based approach to tackle the current pandemic of physical inactivity. There is an urge to include physical inactivity and SB in individuals with CRDs in global action plans. Actions and considerations for future research include harmonizing and promoting PA assessment (especially in low- and middle-income countries) and encompassing the synergistic influences of PA, SB and sleep on health outcomes in CRD populations.

**Table 1 ijerph-19-16986-t001:** Examples of the relevance of chronic respiratory disease for the International Society for Physical Activity and Health’s eight investments for physical activity.

**1** **.** **Whole-of-school programmes**
A life-course approach to PA promotion and the importance of PA as a standalone public health policy [80] is relevant to CRD populations. Whole-of-school programmes and other PA interventions for children are needed before PA habits decline, with further decline along the development of CRDs.
**2** **.** **Active transport**
Air pollution is a major cause of CRDs and CRD-related exacerbations [81,82,83,84,85,86]. The worsening effects of air pollution on respiratory symptoms is a barrier to active transport participation; associated with more severe symptoms and poor health-related quality of life in COPD [73]. Transport in general is often a barrier to PA participation for individuals living with CRDs. In COPD, the benefits of PA outweigh the harms of poor air quality [87,88]. Active transport is an underutilised avenue for increasing PA in CRD populations.
**3** **.** **Active urban design**
In CRDs, the relationship between environmental factors and PA may be influenced by the factors such as feelings of embarrassment caused by being short of breath, coughing or demonstrating challenges with their mobility in public [89]. Sitting is typically interpreted as negative health behaviour, but sitting can also be an enabler to being more physically active [90]. An estimated 90% of people living with COPD in the UK walking at a speed slower than that assumed by pedestrian crossings [91]. Existing active urban design and changes intended to increase population PA [92], such as neighbourhood walkability [93,94], may not always be suitable for people living with CRDs.
**4** **.** **Healthcare**
Pulmonary rehabilitation has PA at its core, through exercise training and commitment to supporting long-term health behaviour change [95]. The benefits of pulmonary rehabilitation in CRD populations, driven largely by PA in the form of exercise, are similar to the benefits of meeting the WHO 2020 PA guidelines [96]. There is a potential synergy between the evidence for pulmonary rehabilitation and the evidence for increasing PA levels in CRD populations. Adjuncts to healthcare services to promote increases in PA may support CRD populations to be more physically active [97].
**5** **.** **Public education, including mass media**
In isolation, interventions targeting public education have had limited success in increasing the population of PA, with the need for tailoring to more specific subgroups, such as CRDs [98]. The recall of PA guidelines by healthcare professionals involved in the care of CRD requires improvement in communication and evaluation [99]. People living with CRDs will need different messaging around PA compared with general adults, but there is currently limited evidence on PA messages in the context of CRDs.
**6** **.** **Sport and recreation for all**
There is a potential for walking sports alternatives to be a safe and enjoyable way to increase PA participation, with some health benefits [100,101]. Whilst there is limited evidence specifically for sports participation within CRD populations, activities such as dancing [102,103], singing [104,105,106,107], Yoga [108] and T’ai Chi [109] are generating a growing interest and evidence base. This area of research has the potential to support a ‘sport and recreation for all’ ethos within the context of CRDs.
**7** **.** **Workplaces**
Although many people living with CRDs such as COPD in HIC are typically older in age and retired, workplace interventions may still offer the potential to increase PA especially in low- and middle-income countries and for people living with certain CRDs such as post-tuberculosis. Evidence for workplace interventions for CRD populations is currently lacking.
**8** **.** **Community-wide programmes**
For people living with COPD, being unable to access community exercise programmes has been reported as a barrier to maintaining PA following pulmonary rehabilitation [110]. Improving the availability, access, and appropriateness of community PA programmes has the potential to benefit people living with CRDs.

## Data Availability

No applicable.

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
