# Peer review of "A Call for Action on Chronic Respiratory Diseases within Physical Activity Policies, Guidelines and Action Plans: Let’s Move!"

_ijerph, 2022, doi:10.3390/ijerph192416986_

Round 1
Reviewer 1 Report
Thank you for your research and submission. The paper is very well written and presented. The topic is also interesting and could have important practical applications. The materials and methods part is very accurate and punctual in the explanations.
In addition, I would like to make a couple of recommendation.
1. To keep the format of the paper. For example, there is a wrong way to quote references in the text. Not the (1), this should be changed to "[1]" .
2. References
There are many areas where the reference description is wrong. For example, spacing!!
3. The important things!
The need to harmonize and promote PA levels (especially low- and middle-income countries) is critical in this study. Also, I think there is a need for empirical research related to this. However, I think this study is only suggesting the need. Therefore, it is hoped that it will be developed into a paper containing specific plans and results related to this.
Author Response
REVIEWER 1
Thank you for your research and submission. The paper is very well written and presented. The topic is also interesting and could have important practical applications. The materials and methods part is very accurate and punctual in the explanations.
Thank you very much for the positive comments on our manuscript.
In addition, I would like to make a couple of recommendation.
- To keep the format of the paper. For example, there is a wrong way to quote references in the text. Not the (1), this should be changed to "[1]"
This has been updated accordingly.
- References
There are many areas where the reference description is wrong. For example, spacing!!
Reference formatting has been corrected and will be further checked during type setting before being published.
- The important things!
The need to harmonize and promote PA levels (especially low- and middle-income countries) is critical in this study. Also, I think there is a need for empirical research related to this. However, I think this study is only suggesting the need. Therefore, it is hoped that it will be developed into a paper containing specific plans and results related to this.
Correct, this paper has been submitted as a commentary to highlight important issues and how they might be addressed. All authors are actively working in this area.
Reviewer 2 Report
This commentary highlights the need for global policies, action plans and specific guidelines in relation to encouraging the uptake of physical activity within individuals who have chronic respiratory disease (CRD). As a researcher with an interest in this chronic condition, I certainly feel this manuscript provides a well thought-out call to action. Overall, I found it an insightful read with some good reasonable suggestions for action. However, there are a few updates and clarifications which need to be made before this article can be considered for publication.
Line 2: Typo… Should be “call” rather than “cal”.
Line 25: Extra gap between “PA” and “in”.
Line 31: I am not totally sure some of the suggested key words are suitable. Perhaps use keywords like “chronic respiratory disease”, “physical activity” and “policy”.
Line 38-45: As many people with CRD are older adults, some reference to the need to complete additional balance exercise is warranted. Also, including an example or two to how the physical activity recommendations have been made specific to disease populations would be useful.
Line 51: These two references only link to COPD. Another couple of references linking to other CRDs would present a fuller picture.
Line 96: Need a reference for this statement.
Line 100-105: “… PA and SB…”. I think it is important to refer to both as I believe changing sedentary behaviour could be a great avenue worth exploring in people with CRD as they are distinct behaviours. Please add “SB” after the other mentions of “PA” in this paragraph.
Line 195-197: I feel this statement could be expanded further, rather than just focusing in on total times in different intensities. Assessing the pattern of activity (e.g. physical activity fragmentation and number of sedentary bouts over 30 minutes), times of day being active / least active (or least sedentary and most sedentary) and free-living physical activity which resembles commonly used exercise tests in CRD (e.g. peak 6-minute cadence values) could also be very important to explore in this population.
Line 261-262: Think you need a reference to highlight poor sleep quality in CRD.
Line 270: I think this should be Ref 68 rather than Ref 69…?
Line 291-293: I think this sentence could be worded a little better.
Box 1: I feel this section contains some great suggestions / clarifications for how CRD can link in with the eight investments in physical activity. I have included some points below to consider:
1. Whole-of-school programmes: I feel you could add something here in relation to targeting children whose relatives have CRDs. I make this point as many children may see their relatives who have CRDs not being active and that may perpetuate through the generations? Although this is just a thought, potentially not feasible but maybe worth considering / adapting…
3. Active urban design: I feel some reference to some of the walkability research could be relevant here and enhance this section (i.e. there are some published papers which focus on enhancing walkability)
4. Healthcare: What about alternative / complementary approaches to encouraging physical activity within the healthcare setting in addition to PR? A review paper I published a while back eludes to this: https://www.tandfonline.com/doi/full/10.3109/15412555.2014.948992 I obviously appreciate there are likely to be more recent reviews on this topic so include what you see as relevant.
Author Response
REVIEWER 2
This commentary highlights the need for global policies, action plans and specific guidelines in relation to encouraging the uptake of physical activity within individuals who have chronic respiratory disease (CRD). As a researcher with an interest in this chronic condition, I certainly feel this manuscript provides a well thought-out call to action. Overall, I found it an insightful read with some good reasonable suggestions for action. However, there are a few updates and clarifications which need to be made before this article can be considered for publication.
Thank you for the kind words and positivity towards our call to action.
Line 2: Typo… Should be “call” rather than “cal”.
Corrected, thank you
Line 25: Extra gap between “PA” and “in”.
Corrected
Line 31: I am not totally sure some of the suggested key words are suitable. Perhaps use keywords like “chronic respiratory disease”, “physical activity” and “policy”.
We have added the following key words:
“Physical activity, sedentary behavior, guidelines, recommendations, health policy, chronic respiratory diseases, low- and middle-income countries”
Line 38-45: As many people with CRD are older adults, some reference to the need to complete additional balance exercise is warranted. Also, including an example or two to how the physical activity recommendations have been made specific to disease populations would be useful.
While we agree with the comment on the importance of balance, we have focussed on the key messages of the WHO guidelines, as this is pertinent to the wider piece. Despite the growing evidence of the benefits of balance exercises on individuals living with CRDs, this kind of exercises is not yet included in the formal PA recommendations from entities such as WHO. The references provided direct readers to the sources describing in detail the WHO physical activity guidelines. We do not seek to critique the approach taken in the development of the guidelines, but to shine a light on the need for evidence in CRD populations, as they are not yet included in the “special populations” recommendations.
Line 51: These two references only link to COPD. Another couple of references linking to other CRDs would present a fuller picture.
We have added three references – one for severe asthma, one for bronchiectasis and one for interstitial lung disease:
https://pubmed.ncbi.nlm.nih.gov/32265311/
https://pubmed.ncbi.nlm.nih.gov/31530430/
https://journals.plos.org/plosone/article?id=10.1371/journal.pone.0277973
Line 96: Need a reference for this statement.
An additional citation of Reference 2 has been added.
Line 100-105: “… PA and SB…”. I think it is important to refer to both as I believe changing sedentary behaviour could be a great avenue worth exploring in people with CRD as they are distinct behaviours. Please add “SB” after the other mentions of “PA” in this paragraph.
We agree with the reviewer and we have added SB throughout the manuscript where appropriate. As the WHO guidelines focus primarily on physical activity (as this is where the best and largest volume of evidence exists), we have reflected this in our considerations.
Line 195-197: I feel this statement could be expanded further, rather than just focusing in on total times in different intensities. Assessing the pattern of activity (e.g. physical activity fragmentation and number of sedentary bouts over 30 minutes), times of day being active / least active (or least sedentary and most sedentary) and free-living physical activity which resembles commonly used exercise tests in CRD (e.g. peak 6-minute cadence values) could also be very important to explore in this population.
We agree with the reviewer that such metrics offer additional behavioural insight. Our ‘Actions and considerations for future research’ reflects the area we believe are most relevant to the WHO guidelines. Since bouts of physical activity were removed from the latest guidelines (previous ≥10 minute bouts of MVPA), prolonged bouts are no longer a requisite for meeting the recommendations. Finally, following the reviewer’s suggestion, we have added the following sentence: “For example, it may be helpful to examine PA intensity in relation to tests of exercise capacity.”
Line 261-262: Think you need a reference to highlight poor sleep quality in CRD.
References 72 and 73 are provided in the preceding sentence, along with other available evidence exploring the interplay between physical activity, sedentary behaviour and/or sleep. Collectively, these support the statement.
Line 270: I think this should be Ref 68 rather than Ref 69…?
Thank you, this has been amended.
Line 291-293: I think this sentence could be worded a little better.
This sentence has been reworded as follows: “The impact of multimorbities and scarce data from LMIC must be addressed for the burden of physical inactivity in CRDs to be included in future global PA guidelines.”
Box 1: I feel this section contains some great suggestions / clarifications for how CRD can link in with the eight investments in physical activity. I have included some points below to consider:
- Whole-of-school programmes: I feel you could add something here in relation to targeting children whose relatives have CRDs. I make this point as many children may see their relatives who have CRDs not being active and that may perpetuate through the generations? Although this is just a thought, potentially not feasible but maybe worth considering / adapting…
This is a very interesting thought and we agree it would warrant further exploring. For the purpose of this table, we feel this may be too specific to be included as a suggestion but we thank the reviewer for sharing their thoughts.
- Active urban design: I feel some reference to some of the walkability research could be relevant here and enhance this section (i.e. there are some published papers which focus on enhancing walkability)
We have added “such as neighbourhood walkability” to this section, with the following references:
https://pubmed.ncbi.nlm.nih.gov/36309283/
https://pubmed.ncbi.nlm.nih.gov/30166322/
- Healthcare: What about alternative / complementary approaches to encouraging physical activity within the healthcare setting in addition to PR? A review paper I published a while back eludes to this: https://www.tandfonline.com/doi/full/10.3109/15412555.2014.948992 I obviously appreciate there are likely to be more recent reviews on this topic so include what you see as relevant.
We agree with the reviewer and we have added a bullet point as follows: “Adjuncts to healthcare services to promote increases in PA may support CRD populations to be more physically active.” The following reference has been included to support the statement:
https://www.cochranelibrary.com/cdsr/doi/10.1002/14651858.CD012626.pub2/full
Reviewer 3 Report
Accept after the comments are incorporated into the submission.

Author Response
REVIEWER 3
Presented suggestions hopefully enhance the submission.
Line 24, 26, etc...… avoid the use of the word ‘we’ in the text. Please reformulate the sentences.
The text has been modified accordingly.
Line 26-30: If you are encompassing synergistic influences of PA, sedentary behavior, and sleep, one paragraph should also be devoted to sleep (measurement, future research, contribution ….) and its influence on CRD.
This relates to the last sentence of the abstract, which summarises our views on future research priorities. The focus of this Commentary is physical activity, in accordance with its focus on WHO guidelines. We discuss sleep in the context of physical activity and capturing the 24-hour picture of behaviour, this believe it does not require a dedicated paragraph in the introduction.
Line 88, 219: Break down the abbreviations HRQoL and GGIR.
HRQoL has been expanded to “health-related quality of life”. GGIR does not have an expanded name as it relates to an R package for accelerometer processing http://cran.r-project.org
Line 249-258: Lifestyle incorporate multiple pillars (PA, nutrition, mental state, environment, sleep, social context …). Why select PA, SB, and sleep as contributing factors to CRD outcome? Please defend its validity over other factors.
These lifestyle behaviours are focus of our commentary and the special issue it has been submitted to. We are not commenting on other lifestyle behaviours as they are beyond the scope of this work.
26 out of 105 references are ten or more years old. Please replace some of them with the latest research.
We believe we have used appropriate references. At the other end, we have 33 references published since 2020. Older references typically relate to findings that have not changed with more recent evidence (e.g. individuals with CRDs are less physically active than healthy controls). "Classic" references do exist in any research field, and are frequently used for being pioneer or perhaps more solid than other (more recent) references. Please let us know if there are specific changes being suggested.
"In summary and next steps," you can introduce the lifestyle coach profession as a vehicle and platform for lifestyle evaluation, mediation, and promotion. I suggest including the holistic view, which can be incorporated into a separate paragraph (its place in non-communicable diseases, future research, measurement tools, and so forth.)
This section has its focus on the WHO guidelines and summarises the issues raised in the Commentary.
Round 2
Reviewer 1 Report
The paper is very well written and presented. The topic is interesting and could have important practical applications. I would like to thank the authors for their various amendments. Also, I would like to highly appreciate that I can develop it into paper. I'm looking forward to it. Thank you.
Reviewer 2 Report
Dear authors. Thank you for providing swift updates to your commentary and feel this has certainly improved the manuscript. I have no further updates to suggest.